# Object2Scene: Putting Objects in Context for Open-Vocabulary 3D Detection

## Abstract

Point cloud-based open-vocabulary 3D object detection aims to detect 3D categories that do not have ground-truth annotations in the training set. It is extremely challenging because of the limited data and annotations (bounding boxes with class labels or text descriptions) of 3D scenes. Previous approaches leverage large-scale richly-annotated image datasets as a bridge between 3D and category semantics but require an extra alignment process between 2D images and 3D points, limiting the open-vocabulary ability of 3D detectors. Instead of leveraging 2D images, we propose *Object2Scene*, the first approach that leverages large-scale large-vocabulary 3D object datasets to augment existing 3D scene datasets for open-vocabulary 3D object detection. *Object2Scene* inserts objects from different sources into 3D scenes to enrich the vocabulary of 3D scene datasets and generates text descriptions for the newly inserted objects. We further introduce a framework that unifies 3D detection and visual grounding, named *L3Det*, and propose a cross-domain category-level contrastive learning approach to mitigate the domain gap between 3D objects from different datasets. Extensive experiments on existing open-vocabulary 3D object detection benchmarks show that *Object2Scene* obtains superior performance over existing methods. We further verify the effectiveness of *Object2Scene* on a new benchmark OV-ScanNet-200, by holding out all rare categories as novel categories not seen during training.

## 1 Introduction

Point cloud-based 3D object detection aims at localizing and recognizing objects from 3D point cloud of scenes. It is a fundamental task in 3D scene perception where remarkable progress has been made in recent years (Zhou et al., 2022b; Liu et al., 2021; Shi et al., 2020; Misra et al., 2021). However, the ability of 3D detection is limited to a small vocabulary due to the limited number of annotated categories in 3D point cloud datasets. As a critical step towards generalizing 3D object detection to real-world scenarios, open-vocabulary 3D object detection aims at detecting categories without annotations in the training set.

Mainstream open-vocabulary 2D detection methods rely on large-scale image-text datasets (Ridnik et al., 2021; Lin et al., 2023) or models pre-trained on these datasets (Gu et al., 2021; Kuo et al., 2022) to provide additional knowledge on the general representations for unseen categories. Unfortunately, similar point-text data pairs are limited due to the expenses of 3D annotations and point cloud collection, making those 2D approaches not directly applicable to the 3D scenario. Previous approaches (Lu et al., 2023; 2022) leverage richly annotated image datasets as a bridge and transfer knowledge from images to 3D point clouds to address this challenge. However, these images and point clouds typically do not have explicit correspondences, which casts a severe challenge for 2D-3D alignment and hinders the open-vocabulary ability of 3D detectors. Besides, the generated pseudo 3D bounding boxes based on 2D detectors are sub-optimal and limit the open-vocabulary 3D detection capability in the localization stage.

The recently proposed large-scale 3D object datasets (Chang et al., 2015; Uy et al., 2019; Deitke et al., 2022) pave the way for learning large-vocabulary 3D object representations without 2D data. These 3D object datasets have been used in recent 3D recognition methods (Zhang et al., 2022b; Zhu et al., 2022; Xue et al., 2023), achieving astonishing open-vocabulary point cloud recognition ability. However, these methods can not address the issue of localizing objects required in 3D

detection. Different from 2D images, a 3D object could be easily inserted into a point cloud scene accompanied by naturally precise 3D location and box annotation. Therefore, we raise the question: *can we train the models to learn open-vocabulary 3D object detection by leveraging 3D object datasets to augment 3D scene datasets?* The 3D scene datasets are foundational for 3D object detection with limited categories, while the 3D object datasets can act as a database of 3D objects to largely enrich the vocabulary size of the 3D detector.

An intuitive augmentation approach could be putting the large-scale 3D objects in the 3D context to broaden the vocabulary size. Based on the augmented context, the 3D detector can access large-vocabulary 3D objects with precise 3D location annotation. However, such a naive 3D object insertion approach leads to an *inconsistent annotation issue*, where the unseen category objects from the original scene are not annotated, but the unseen category objects that are newly inserted into the scene are annotated. To this end, we propose **Object2Scene**, which puts objects into the 3D context to enrich the vocabulary of 3D scene datasets and generates text prompts for the newly inserted objects. Specifically, we choose seen objects in the scene as reference objects to guide the physically reasonable 3D object insertion according to the spatial proximity illustrated in 3.2. Besides, to mitigate this inconsistent annotation issue, we introduce language grounding prompts to diminish the ambiguity of class label annotations. The generated grounding prompts such as "a table that is near a plant that is at the room center" could eliminate the ambiguity and refer to the specific inserted table. By such construction, we can achieve scene-level 3D object-text alignment in a large vocabulary space.

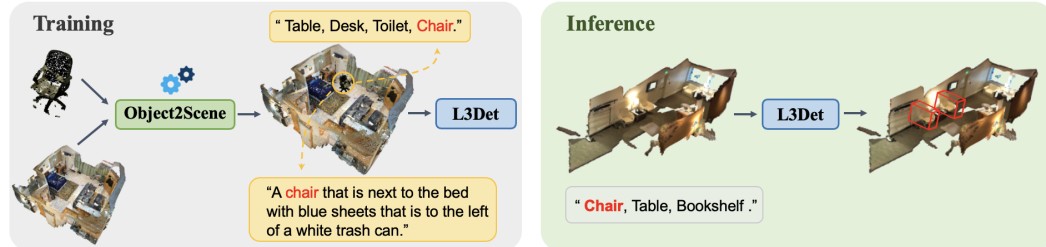

Figure 1: By utilizing the 3D object datasets, Object2Scene empowers the 3D detector (L3Det) with strong open-vocabulary capability. The training process is shown on the left, where Object2Scene generates training data for L2Det by inserting unseen objects into the 3D scene and generating grounding prompts for the inserted objects. The inference process is shown on the right.

Furthermore, we explore the model design of open-vocabulary 3D object detection with Object2Scene. To adapt the model for both object category annotations and language grounding prompts, we propose a unified model, named **Language-grounded 3D point-cloud Detection (L3Det)**, for 3D detection and 3D language grounding. L3Det takes the point cloud of a 3D scene and a text prompt (the text prompt for object detection is a list of object categories) as input, and grounds the prompt to object bounding boxes with a transformer-based encoder-decoder architecture. Since we mainly align the 3D objects from 3D object datasets with the text in the large vocabulary space, another challenge of Object2Scene is *the domain gaps between inserted objects from 3D object datasets and existing objects in the original 3D scene datasets*. We further introduce a **Cross-domain Category-level Contrastive Learning** approach to mitigate such domain gaps. Specifically, the contrastive loss specifically aims to bring feature representations of same-category objects closer together and push feature representations of different-category objects farther apart, regardless of the source datasets. The proposed cross-domain category-level contrastive learning method further reduces the domain gap between feature representations of the inserted 3D objects and the original 3D scenes, and guides the model towards learning source-agnostic generalizable 3D representations.

Our contributions are summarized as follows. 1) We propose Object2Scene, the first approach that leverages large-scale large-vocabulary 3D object datasets and existing 3D scene datasets for open-vocabulary 3D object detection, by inserting 3D objects into 3D scenes and generating language grounding prompts for the inserted objects. 2) We introduce a language-grounded 3D detection framework L3Det to achieve 3D object-text alignment and propose a novel cross-domain category-level contrastive learning method to mitigate the domain gap between inserted 3D objects and original 3D scenes in Object2Scene. 3) We evaluate Object2Scene on open-vocabulary 3D object de-

tection benchmarks on ScanNet and SUN RGB-D, and validate the effectiveness of our proposed approach with extensive experiments.

## 2 RELATED WORK

### 2.1 3D OBJECT DATASETS

Current 3D object datasets can be divided into two categories: synthetic and real-scanned. For the synthetic CAD model datasets, ShapeNet (Chang et al., 2015) has 51300 3D CAD models in 55 categories, and ModelNet40 (Wu et al., 2015) has 12311 3D CAD models in 40 categories. Recently, a large-scale 3D CAD model dataset, Objaverse (Deitke et al., 2022), consisting of 818k objects and 21k categories, is proposed to enable research in a wide range of areas across computer vision. For the real-scanned 3D object datasets, ScanObjectNN (Uy et al., 2019) is a real-world point cloud object dataset based on scanned indoor scenes, containing around 15k objects in 15 categories. OmniObject3D (Wu et al., 2023b) includes 6k 3D objects in 190 large-vocabulary categories, sharing common classes with popular 2D and 3D datasets. Our method validates that these various 3D object datasets with large vocabulary categories could greatly benefit open-vocabulary 3D object detection.

### 2.2 MIXED SAMPLE DATA AUGMENTATION

Mixed sample is a common data augmentation strategy usually can increase the number of diversity of the training dataset and better 3D reprenstation learning. PointCutmix (Zhang et al., 2022a) generates new training data by replacing the points in one sample with their optimal assigned pairs for robust 3D object recognition. Mix3D (Nekrasov et al., 2021) takes the union over the two point cloud scenes to achieve scene-level data augmentation. Besides, RandomRooms (Rao et al., 2021) and BackToReality (Xu et al., 2022) place the 3D objects following the basic physical constraints to construct the virtual scenes for better scene understanding. PointRCNN (Shi et al., 2019) copies the 3D objects and pastes them into the scene data to generate mixed reality outdoor scenes. Differently, out method inserts the large-vocabulary 3D objects from various object datasets in an anchor-guided manner and generate appropriate language prompts to further guide detectors with open-vocabulary detection capability.

### 2.3 OPEN-VOCABULARY OBJECT DETECTION

Open-vocabulary object detection aims at detecting the categories that are not provided bounding box labels during training (Zareian et al., 2021), which first rises in 2D domain due to the large amount of image-text and region-text pairs (Schuhmann et al., 2022; Sharma et al., 2018). Most methods either directly use large-scale image-text pairs (Lin et al., 2023; Zhou et al., 2022a) to provide weak supervision signals when training the detectors or adopt the vision-language models (Radford et al., 2021) pre-trained on large-scale image-text pairs by distillation (Gu et al., 2021; Wu et al., 2023a), fine-tuning (Zhong et al., 2022), or freezing (Kuo et al., 2022). In 3D domain, utilizing the corresponding 2D images of the point cloud scene, recent approaches (Lu et al., 2023; 2022) first use pseudo 3D bounding boxes in training to obtain the localization ability and then connect the point cloud with the image by CLIP (Radford et al., 2021) or image classification dataset (Deng et al., 2009) to empower the 3D detector open-vocabulary recognition ability. Our method directly uses large-scale large-vocabulary 3D object datasets to achieve both open-vocabulary 3D localization and recognition simultaneously in an end-to-end manner, alleviating the need of corresponding 2D images.

### 2.4 3D REFERENTIAL LANGUAGE GROUNDING

Conventional 3D visual grounding methods (Zhao et al., 2021; Achlioptas et al., 2020; Feng et al., 2021; He et al., 2021; Huang et al., 2021; Roh et al., 2022; Yuan et al., 2021) adopt a 'detect-then-match' paradigm. These methods first obtain the text features and object proposals by a pre-trained language model and a 3D detector, respectively, then learn to match the object and text features in training and select the best-matched object for each concept in inference. The recent rising one-stage methods (Jain et al., 2022; Wu et al., 2022; Luo et al., 2022) tend to fuse the text feature into the process of point-cloud feature extraction and detect the text-conditioned object directly, which tends

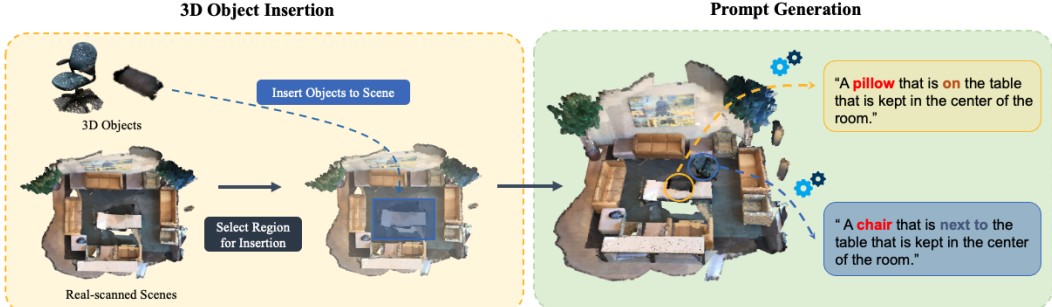

Figure 2: Overall pipeline for Object2Scene. The objects are sampled from 3D object datasets and inserted into the real-scanned scene. Then we generate grounding prompts for the inserted objects.

to produce better results. Our proposed L3Det further simplifies the one-stage detector architecture and is generic to language-guided 3D object detection.

## 3   OBJECT2SCENE

To tackle open-vocabulary 3D object detection with the limited annotations in existing 3D scene datasets, we propose **Object2Scene**. As shown in Figure 2, Object2Scene first inserts 3D objects from large-scale large-vocabulary 3D object datasets (Chang et al., 2015; Uy et al., 2019; Deitke et al., 2022) into 3D scenes to enrich the vocabulary size (Section 3.1), and then generates grounding prompts for the inserted objects (Section 3.2).

### 3.1   ANCHOR-GUIDED 3D OBJECT INSERTION

The key to empowering the 3D object detector with open-vocabulary abilities is to relate the 3D object representations with large-vocabulary text. Unlike 2D methods that can take advantage of large amounts of image-text pairs, large-scale 3D point-cloud-caption pairs for point cloud scenes are infeasible, while 3D object datasets are more economical and tend to include large vocabulary sizes. Thus, we introduce large-scale and large-vocabulary 3D object datasets into the 3D scenes. However, simple random insertion of objects into the scenes often disrupts the coherence of the scene and makes it hard to generate text descriptions. Since the seen objects annotations are accessible, these objects can act as reference objects (anchors) to guide the object insertion, so we propose a physically reasonable insertion approach: **Anchor-guided 3D Object Insertion** with three steps: Anchor and Object Selection, Object Normalization and Resampling, and Object Placement.

**Object and Anchor Selection.** We randomly choose one object from the seen objects in the scene as the anchor, and then we randomly sample one object as the target object from the 3D object datasets. The target object comes from a large vocabulary size and may belong to seen or unseen categories except the same category as the anchor.

**Object Normalization and Resampling.** Due to the different scanning devices and different approaches for data collection and pre-processing, there exists the domain gap of point cloud distributions among different 3D datasets. We predefined a collection of similar objects categories. For the target object, if there exist similar categories in the seen categories, we first normalize the scale of the target object to the average size of the similar category objects and then resample the point cloud to match the average number of points.

**Object Placement.** In order to place objects in a physically reasonable way, we divide objects into three types, stander, supporter, and supportee, following Xu et al. (2022). Standers are objects that can only be supported by the ground and cannot support other objects. Supporters are objects that can only be supported by the ground and can support other objects. Supportees are objects that supporters can support. We place the target in potentially physically valid locations around the anchor. Specifically, a rectangular region centered at the anchor location $(x_a, y_a)$ is determined, and then we compute a $z$ axis height map of the insertion region and iteratively sample centroid coordinates candidates where the target can be placed. Then we will check whether the placement is physically reasonable according to 1) the categories that the anchor and target belong to (stander,

supporter, or supportee), and 2) whether the inserted target would collide with existing objects in the scene.

## 3.2 Object Grounding Prompt Generation

The 3D object insertion approach creates new 3D scenes by combining original scenes that have been annotated with seen categories, along with new objects from large vocabulary categories that include seen and unseen ones. A straightforward way of deriving detection annotations is augmenting the existing object detection annotations of the seen classes with the bounding boxes and class labels of the inserted objects from large-vocabulary 3D datasets, and directly training the detector on the augmented 3D detection datasets. However, such naive label assignment mechanism brings up the inconsistent annotation problem mentioned in Section 1. Such inconsistencies may cause ambiguity in annotations and hinder the training of the open-vocabulary 3D detection models in the regular detection training paradigm. Therefore, we propose to generate language grounding prompts to reduce the ambiguity of category-level annotations and to provide clear training guidance for the models.

Similar to SR3D (Achlioptas et al., 2020), we can generate the spatial prompt for the inserted target object according to the following template: $\langle target \rangle \langle spatial\ relationship \rangle \langle anchor \rangle$, e.g. *"the table that is next to the plant"*. The spatial relationship can be categorized into three types: Horizontal Proximity, Vertical Proximity, and Allocentric, according to the location of the anchor and target and the intrinsic self-orientation of the anchor. More details can be found in our appendix. However, the generated simple spatial prompt sometimes may not be strong enough to distinguish the target from other same-category objects in the scene. To further diminish the ambiguity of annotations, we combine the generated spatial utterance and the ground-truth referring expression of the anchor together and generate the **Relative Location Prompt**. For example, if the reference object when inserting the table is a plant with a ground-truth referring expression grounding annotation "a plant that is at the room center", we can generate the prompt for the newly inserted table as "a table that is next to a plant that is at the room center". It is guaranteed that the generated prompt uniquely refers to the table because the original grounding prompt uniquely refers to the plant.

Besides, we also propose **Absolute Location Prompt**. This type of prompt is constructed solely based on the object's position in the scene, rather than relying on other objects. For example: "a table that is closer to the center/corner/wall of the room".

## 4 Open-vocabulary 3D Object Detection with Object2Scene

In order to train an open-vocabulary 3D object detector with the proposed Object2Scene, we propose a new, simple, but strong baseline that unifies 3D detection and 3D visual grounding, named **L3Det** (Language-grounded 3D object detection). L3Det enables training with both detection prompts and more accurate grounding prompts introduced in Section 3. We leverage the 3D scenes and text prompts generated by Object2Scene for training. To mitigate the domain gap between the feature representations of inserted objects and existing objects in the original scene, we propose a **cross-domain category-level contrastive learning** approach to force source-agnostic generalizable feature representations for the multiple-source 3D objects in the 3D scenes.

### 4.1 L3Det: Langauge-grounded 3D Object Detection

**Model Architecture.** Our L3Det model follows the generic transformer-based design with the encoder-decoder architecture, as shown in the left side of Figure 3. The input to the model is a point cloud and a text prompt. We extract the point visual tokens $\mathbf{V} \in \mathbb{R}^{n \times d}$ with the PointNet++ encoder and text queries $\mathbf{T} \in \mathbb{R}^{l \times d}$ with the pre-trained RoBERTa text encoder, where $l = 256$ is the maximum length of the text. Following BUTD-DETR (Jain et al., 2022), the top-$K$ highest scoring visual tokens are fed into an MLP to predict the non-parametric object queries, which are updated iteratively through $N$ decoder layers. In each decoder layer, the text queries and non-parametric object queries attend to the point visual features $\mathbf{V}$ with cross-attention to gather the visual information, followed by a self-attention among non-parametric queries and text queries. Finally, the prediction head takes the updated object queries as input and predicts the 3D boxes and object features for aligning the predicted objects with text tokens.

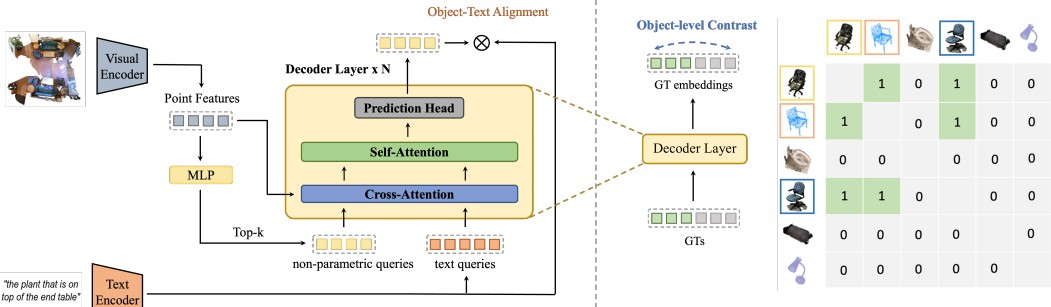

Figure 3: Open-vocabulary 3D object detection with Object2Scene. The figure on the left-hand side shows the model architecture of L3Det. The figure on the right-hand side shows the cross-domain category-level contrastive learning approach. Given 6 objects illustrated in the figure, the contrastive loss brings together (denoted by "1" in the matrix) the features of the three objects belonging to the category "chair", despite the fact that they are from different source datasets. The object features of different categories are pushed away from each other (denoted by "0" in the matrix).

**Training Supervision.** The overall training supervision is the summation of three loss terms: localization loss $\mathcal{L}_{loc}$, alignment loss $\mathcal{L}_{align}$, and our proposed cross-modal category-level contrastive loss $\mathcal{L}_{cl}$ (introduced in Section 4.2). The localization loss $\mathcal{L}_{loc}$ is a combination of L1 and generalized IoU (Rezatofighi et al., 2019) losses between predicted and ground-truth bounding boxes. Following GLIP (Li et al., 2022), the alignment loss measures the alignment between predicted object features and the text queries. Specifically, we compute the object-text alignment scores by $\mathbf{S}_{align} = \mathbf{P}\mathbf{T}^T$, where $\mathbf{P} \in \mathbb{R}^{N \times d}$ is the object features predicted by the prediction head and $\mathbf{T} \in \mathbb{R}^{l \times d}$ is the text queries extracted by the pretrained text encoder. We calculate the target alignment score matrix $\mathbf{S}_{target} \in \{0, 1\}^{N \times M}$ based on the ground-truth alignment between box locations and text query tokens. Binary sigmoid loss (Li et al., 2022) between $\mathbf{S}_{align}$ and $\mathbf{S}_{target}$ is adopted for the alignment loss.

**Detection Prompts Supervision.** Following BUTD-DETR (Jain et al., 2022), we empower the L3Det general detection ability with detection prompts comprised of a list of object category labels. For the detection prompts, the task is to identify and locate all objects belonging to the category labels mentioned in the prompt, if they are present in the scene.

## 4.2 CROSS-DOMAIN CATEGORY-LEVEL CONTRASTIVE LEARNING

As demonstrated in Section 3, our Object2Scene approach inserts 3D objects into 3D scenes to generate enriched scenes for training. A consequent problem is the domain gap between the newly inserted objects and existing objects in the original scene. Since the newly inserted objects are from 3D object datasets that have different data distributions from the 3D scene datasets, causing severe misalignment between their 3D feature representations. To mitigate this challenge, we propose a cross-domain category-level contrastive learning approach that leverages contrastive loss on cross-domain objects to learn source-agnostic 3D feature representations. Since the objects we inserted cover both seen and unseen categories, we can construct positive pairs using the seen objects from different datasets. For example, for seen class *desk*, one constructed positive pair could consist of one *desk* from the existing scene and one *desk* from the other 3D object datasets, and the model is trained to minimize the distance between these two same category objects from different datasets. In this way, the model can learn the general representation of the same seen class objects from different datasets, thereby implicitly improving its generalization ability on unseen classes.

As shown in the right side of Figure 3, let $f_i$ denote the object feature for the $i$-th object in the mini-batch extracted from a decoder layer. In the mini-batch, objects with the same class labels form positive pairs for contrastive learning and objects with different class labels form negative pairs. The cross-domain category-level contrastive learning loss can be represented as:

$$\mathcal{L}_{cl} = -\frac{1}{N} \sum_{i=1}^{N} \log \frac{\sum_{i \neq j, y_i = y_j, j=0,\cdots,N} \exp(f_i^\top f_j / \tau)}{\sum_{i \neq k, k=0,\cdots,N} \exp(f_i^\top f_k / \tau)}, \quad (1)$$

where $N$ is the number of objects in the mini-batch, $y_i$ denotes the class label of the $i$-th object, and $\tau$ is a temperature hyper-parameter. The contrastive loss is also applied to the features of each decoder layer, similar to the detection losses. It is worth noting that the above loss is a generalized contrastive loss that allows multiple positive samples. With the proposed cross-domain category-level contrastive loss, the model is forced to learn generalizable feature representations based on their class labels irrespective of the source dataset of the objects.

## 5 EXPERIMENT

In this section, we introduce three benchmarks for open-vocabulary 3D object detection, and conduct extensive experiments and analysis to validate the effectiveness of our proposed Object2Scene.

### 5.1 BENCHMARKS

**OV-ScanNet20.** Following OV-3DETIC (Lu et al., 2022), we split 20 categories from ScanNet into 10 seen classes (bathtub, fridge, desk, night stand, counter, door, curtain, box, lamp, bag) and 10 unseen classes (toilet, bed, chair, sofa, dresser, table, cabinet, bookshelf, pillow, sink).

**OV-SUN RGB-D20.** We split 20 categories from SUN RGB-D datasets into 10 seen classes (table, night stand, cabinet, counter, garbage bin, bookshelf, pillow, microwave, sink, and stool) and 10 unseen classes (toilet, bed, chair, bathtub, sofa, dresser, scanner, fridge, lamp, and desk).

**OV-ScanNet200.** To fully verify the open-vocabulary ability on a larger number of categories, we resort to the large vocabulary of ScanNet200 (Rozenberszki et al., 2022) where the 200 categories are split into head (66 categoreis), common (68 categories) and tail (66 categories), based on the frequency of number of labeled surface points in the training set. To achieve an open-vocabulary setting, we define the head categories as seen classes, and common and tail categories as unseen classes.

The evaluation metrics for open-vocabulary object detection are Average Precision (AP) and mean Average Precision (mAP) at IoU thresholds of 0.25, denoted as $AP_{25}$, $mAP_{25}$, respectively. To test the open-vocabulary abilities, results are tested on the unseen categories.

### 5.2 IMPLEMNTATION DETAILS

Three commonly-used 3D object datasets are chosen as the data source for Object2Scene: (1) ShapeNet (Chang et al., 2015) with 55 categories and 51300 objects, (2) OmniObject3D (Uy et al., 2019) with 190 categories and 6k objects, and (3) Objaverse (Deitke et al., 2022) with 21k categories and 818k objects. We introduce objects from ShapeNet and OmniObject3D for the OV-ScanNet20 and OV-SUN RGB-D20 benchmark and additionally Objaverse for the OV-ScanNet200 benchmark.

During training, we use the class labels to form the detection prompts. Specifically, OV-ScanNet20 and OV-SUN RGB-D20, we generate the detection prompts by sequencing the 20 class names. For OV-ScanNet200, we randomly sample 20 categories to generate the detection prompts for each training iteration. For Relative Location Prompt generation, and for SUN RGB-D dataset, we generate synthetic referring express generation, we choose ScanRefer (Chen et al., 2020) for ScanNet dataset as referring expression grounding annotations, and for SUN RGB-D dataset, we generate synthetic referring expression grounding annotations following SR3D generation process (Achlioptas et al., 2020).

### 5.3 MAIN RESULTS

We report our results on three benchmarks and compare them with previous methods on OV-ScanNet20 and OV-SUN RGB-D20 benchmarks. We compare our proposed approach with previous approaches OV-PointCLIP (Zhang et al., 2022b), OV-Image2Point (Xu et al., 2021), Detic-ModelNet (Zhou et al., 2022a), Detic-ImageNet (Zhou et al., 2022a), and OV-3DETIC (Lu et al., 2022) introduced in (Lu et al., 2022). The results are presented in Table 1 and Table 2. The results reveal that our Object2Scene, by resorting to the 3D object datasets to enable open-vocabulary 3D detection, outperforms previous state-of-the-art by 12.56% and 23.16% in $mAP_{25}$ on OV-ScanNet20 and OV-SUN RGB-D20, respectively. OV-PointCLIP and Detic-ModelNet obtain open-vocabulary

Table 1: Detection results ($AP_{25}$) on unseen classes of OV-ScanNet20.

| Methods | toilet | bed | chair | sofa | dresser | table | cabinet | bookshelf | pillow | sink | mean |
|---|---|---|---|---|---|---|---|---|---|---|---|
| OV-PointCLIP Zhang et al. (2022b) | 6.55 | 2.29 | 6.31 | 3.88 | 0.66 | 7.17 | 0.68 | 2.05 | 0.55 | 0.79 | 3.09 |
| OV-Image2Point Xu et al. (2021) | 0.24 | 0.77 | 0.96 | 1.39 | 0.24 | 2.82 | 0.95 | 0.91 | 0.00 | 0.08 | 0.84 |
| Detic-ModelNet Zhou et al. (2022a) | 4.25 | 0.98 | 4.56 | 1.20 | 0.21 | 3.21 | 0.56 | 1.25 | 0.00 | 0.65 | 1.69 |
| Detic-ImageNet Zhou et al. (2022a) | 0.04 | 0.01 | 0.16 | 0.01 | 0.52 | 1.79 | 0.54 | 0.28 | 0.04 | 0.70 | 0.41 |
| OV-3DETIC Lu et al. (2022) | 48.99 | 2.63 | 7.27 | 18.64 | 2.77 | 14.34 | 2.35 | 4.54 | 3.93 | 21.08 | 12.65 |
| L3Det | **56.34** | **36.15** | **16.12** | **23.02** | **8.13** | **23.12** | **14.73** | **17.27** | **23.44** | **27.94** | **24.62** |

Table 2: Detection results ($AP_{25}$) on unseen classes of OV-SUN RGB-D20.

| Methods | toilet | bed | chair | bathtab | sofa | dresser | scanner | fridge | lamp | desk | mean |
|---|---|---|---|---|---|---|---|---|---|---|---|
| OV-PointCLIP Zhang et al. (2022b) | 7.90 | 2.84 | 3.28 | 0.14 | 1.18 | 0.39 | 0.14 | 0.98 | 0.31 | 5.46 | 2.26 |
| OV-Image2Point Xu et al. (2021) | 2.14 | 0.09 | 3.25 | 0.01 | 0.15 | 0.55 | 0.04 | 0.27 | 0.02 | 5.48 | 1.20 |
| Detic-ModelNet Zhou et al. (2022a) | 3.56 | 1.25 | 2.98 | 0.02 | 1.02 | 0.42 | 0.03 | 0.63 | 0.12 | 5.13 | 1.52 |
| Detic-ImageNet Zhou et al. (2022a) | 0.01 | 0.02 | 0.19 | 0.00 | 0.00 | 1.19 | 0.23 | 0.19 | 0.00 | 7.23 | 0.91 |
| OV-3DETIC Lu et al. (2022) | **43.97** | 6.17 | 0.89 | **45.75** | 2.26 | 8.22 | 0.02 | 8.32 | 0.07 | 14.60 | 13.03 |
| L3Det | 34.34 | **54.31** | **29.84** | **51.65** | **34.12** | **17.12** | **5.23** | **13.87** | **11.40** | **15.32** | **25.42** |

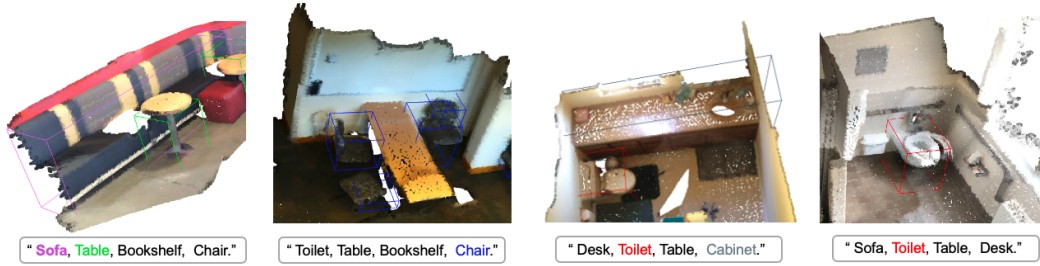

Figure 4: Qualitative results for open-vocabulary 3D object detection results. For each scene, the detection prompt is shown under the input point cloud. The colors of bounding boxes correspond to the classes in the prompts.

abilities from ModelNet, but both approaches perform poorly due to the large domain gap between the CAD 3D models and point clouds of ScanNet and SUN RGB-D obtained from RGB-D sensors. OV-Image2Point and Detic-ImageNet try to directly transfer the knowledge from image to point cloud which ignores the modality gap between 2D and 3D. Our proposed Objecr2Scene approach and L3Det with cross-domain category-level contrastive loss alleviate the aforementioned issues of previous approaches and achieve state-of-the-art performance. We also test our approach on OV-ScanNet200 benchmark, and the $mAP_{25}$ on unseen categories is 10.1% for common categories and 3.4% for tail categories. We illustrate the qualitative results on ScanNet dataset in Figure 4.

## 5.4 ABLATION STUDY

We conduct ablation studies on the OV-ScanNet20 benchmark to justify the contributions of each component and compare various design choices.

**Diversity of the inserted 3D objects** We investigate whether introducing diverse 3D objects from multiple 3D object datasets helps the model learn better 3D representations for open-vocabulary 3D object detection. For the OV-ScanNet20 benchmark, we experiment with inserting objects from ShapeNet only, from OmniObject3D only, and from both ShapeNet and OmniObject3D, respectively. Evaluation results are shown in Table 3a, where the "overlap $mAP_{25}$" refers to the $mAP_{25}$ of categories that can retrieve object instances from both ShapeNet and OmniObject3D datasets, and "$mAP_{25}$" refers to the commonly-defined $mAP_{25}$ averaged over all unseen classes. The ablation study results demonstrate the effectiveness of introducing more diverse objects by leveraging multiple datasets from different sources.

**Randomness in 3D object placement** We explore whether the randomness introduced in 3D object placement is essential for our approach, *i.e.*, what if the location to insert each object for each scene is fixed without any randomness? To avoid the distraction from the randomness of object selection, we fix the selected objects for each scene. For 3D object placement without randomness, we pre-generate the scenes with inserted objects offline before training starts and fix those generated scenes

Table 3: Ablation study on 3D object diversity and generated text prompts.

(a) Ablation study on 3D object diversity.

| 3D object dataset | overlap $mAP_{25}$ | $mAP_{25}$ |
|---|---|---|
| ShapeNet | 7.87 | 11.21 |
| OmniObject3D | 11.31 | 16.92 |
| ShapeNet + OmniObject3D | 15.67 | 24.62 |

(b) Ablation study on the generated text prompts.

| Prompt type | $mAP_{25}$ |
|---|---|
| Detection Prompt | 14.91 |
| Absolute Location Prompt | 17.42 |
| Relative Location Prompt | 21.31 |

Table 4: Ablation study on object normalization and resampling, data augmentation, and cross-domain object-level contrastive learning.

| Norm, Resample | Augmentation | Contrastive | $mAP_{25}$ |
|---|---|---|---|
| | | | 2.34 |
| ✓ | | | 6.41 |
| ✓ | ✓ | | 7.34 |
| ✓ | ✓ | ✓ | 11.21 |

for each epoch. For 3D object placement with randomness, we adopt the same selected 3D objects for insertion but generate the scenes online and introduce randomness in deciding where to place the selected objects. The results show that the randomness in object placement boosts the performance from 18.48% $mAP_{25}$ to 21.31% $mAP_{25}$, bringing 2.83% $mAP_{25}$ gain.

**Language prompts** As introduced in Section 3, the language prompts are critical for training the unified 3D detection and grounding model. We design three types of prompts, namely detection prompts, absolute location prompts, and relative location prompts. We compare the effectiveness of those three types of prompts for learning the feature representations for unseen classes. Results shown in Table 3 indicate that the relative location prompts are the most effective language prompts for open-vocabulary 3D detection. This observation aligns with our assumption that the relative location prompts eliminate the inconsistent annotation issue of objects from the unseen categories.

**Object normalization and resampling** In Section 3.1, we introduce the four steps of inserting a 3D object into a 3D scene, where the second step is normalizing and resampling the 3D object according to the reference point cloud distribution from the 3D scene dataset. This normalization and resampling step is essential for reducing the domain gap between different datasets. As shown in Table 4, the object normalization and resampling increases $mAP_{25}$ from 2.34% to 6.41%.

**Data augmentation** We apply commonly-used object point cloud augmentation techniques including rotation, point dropping, and jittering for training the object detector. The ablation in Table 4 illustrates that applying the data augmentation further increases $mAP_{25}$ from 6.41% to 7.34%.

**Cross-domain category-level contrastive learning** We introduce cross-domain category-level contrastive learning to reduce the domain gap between inserted objects from 3D object datasets and existing objects from the original 3D scene datasets. We justify its effectiveness by an ablation study shown in Table 4. In the ablation setup of Table 4, only ShapeNet is regarded as external 3D object dataset to augment the 3D scene dataset. It is shown that the cross-domain category-level contrastive learning method boosts the $mAP_{25}$ of open-vocabulary 3D detection from 7.34% to 11.21%.

# 6 CONCLUSION

We propose *Object2Scene*, which adopts large-scale large-vocabulary 3D object datasets to enrich the vocabulary of 3D scene datasets for open-vocabulary 3D object detection, by physically resonable inserting 3D objects into 3D scenes and generating language grounding prompts for the inserted objects. We further introduce a unified model, *L3Det*, for 3D detection and grounding, with a cross-domain category-level contrastive loss to bridge the domain gap between inserted objects and original scenes. Extensive experiments and ablations on open-vocabulary 3D detection benchmarks validate the effectiveness of our approach. We believe that our attempt will shed light on future research in open-vocabulary 3D object detection and the broader field of open-world 3D perception.

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

# A EXPERIMENT DETAILS

This section provides more implementation details of the proposed Object2Scene approach and L3Det model.

**Pseudo Code of Object Placement**   Here we provide pseudo code for Object Placement in Object2Scene as a reference.

---
**Algorithm 1** Object Placement Pseudo Code

---
1: Given target object $A$, anchor object $B$ and Scene height map $H$
2: $iter\_num \leftarrow 0$
3: $insert\_area \leftarrow [B.x_{min} - 1, B.x_{max} + 1, B.y_{min} - 1, B.y_{max} + 1]$
4: **while** $iter\_num \leq max\_instance\_placing\_iterations$ **do**
5:    $(x, y) \leftarrow$ RANDOMSAMPLE($insert\_area$)
5:    **if** $(x, y)$ is in the $B's$ object box area **then**
5:       **if** ISVALIDSUPPORT($A.type, B.type$) **then**
5:          $centroid \leftarrow (x, y, H[x, y] + A.height/2)$
5:          $C \leftarrow$ OBJECT($A.size, centroid$)
5:          $available \leftarrow$ CHECKCOLLISION($C, Scene$)
5:       **end if**
5:    **else**
5:       $centroid \leftarrow (x, y, H[x, y] + A.height/2)$
5:       $C \leftarrow$ OBJECT($A.size, centroid$)
5:       $available \leftarrow$ CHECKCOLLISION($C, Scene$)
5:    **end if**
5:    **if** $available$ **then**
5:       **break**
5:    **end if**
6:    $iter\_num + = 1$
7: **end while**=0

---

**TSNE visualization of the Contrastive Learning Loss**   We supplement more visual results here to validate the effectiveness of our Cross-domain Category-Level Contrastive Learning. We show the TSNE visualization in Figure 5. It can be very obvious to see that object features of the same category in different datasets are closer.

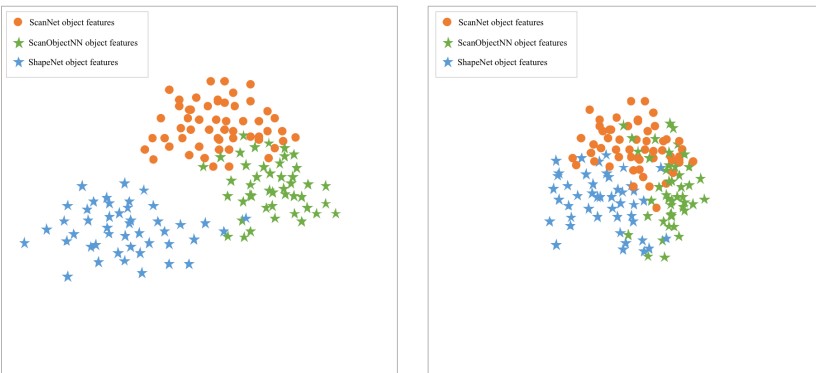

Figure 5: TSNE visualization to analyze the influence of adding the Cross-Domain Object-level Contrastive Learning.

**Prompt Generation for Object2Scene**   Here we provide detailed information for the grounding prompt generation process introduced in Section 3.2. Following SR3D, we generate the spatial prompt for the inserted target object using the following template: $\langle target - class \rangle \langle spatial - relationship \rangle \langle anchor - class \rangle$. Since the target and anchor classes are determined during 3D

Table 5: Performance change in the architecture modification from BUTD-DETR to L3Det.

| Method | Accuracy |
| --- | --- |
| BUTD-DETR | 52.2 |
| + remove box stream | 51.0 |
| + with concatenated Visual and Language Streams | 50.1 |
| + remove cross-encoder | 47.2 |
| + replace with our L3Det decoder (using parallel text and object query | 51.3 |
| + replace with GLIP alignment loss (Our L3Det) | 52.8 |

object insertion, we need to describe the spatial relationship according to their relative locations. we divide the spatial object-to-object relations into three categories:

1. **Vertical Proximity:** It indicates the target is *on* the anchor object.
2. **Horizontal Proximity:** This indicates the target is around the anchor object and can be represented by words like: *next to* or *close to*.
3. **Allocentric:** Allocentric relations are actually based on Horizontal Proximity, which encodes information about the location of the target with respect to the self-orientation of the anchor, which can be represented by words like: *left*, *right*, *front*, *back*.

Once we obtain the generated spatial prompt such as *"the table that is next to the bar stool."*, given a text sentence of anchor object *"it is a wood bar stool. The stool is in the kitchen at the bar. It is the very first stool at the bar."* in ScanRefer (Chen et al., 2020), we utilize the off-the-shelf tool to decouple the text and obtain the main object, auxiliary object, attributes, pronoun, and relationship of the sentence as shown in Figure 6, following EDA (Wu et al., 2022). Then we replace the main object in the sentence with the inserted target object and the original main object becomes the auxiliary object. Combining the spatial prompt, the generated grounding prompt could be *"it is a table next to a wood bar stool. The stool is in the kitchen at the bar. The stool is the very first stool at the bar."*

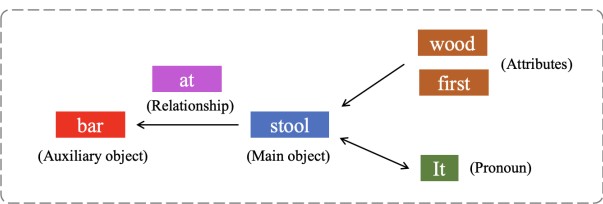

Figure 6: Sentence decoupling illustration.

**Details of the model architecture of L3Det**    In our proposed model L3Det, the point cloud feature tokens $\mathbf{V} \in \mathbb{R}^{n \times d}$ are extracted by PointNet++ (Qi et al., 2017) encoder pre-trained on ScanNet (Dai et al., 2017) seen classes, where $n = 1024$ denotes the number of input points. The text query tokens $\mathbf{T} \in \mathbb{R}^{l \times d}$ are extracted by the pre-trained RoBERTa (Liu et al., 2019) text encoder, where $l = 256$ is the maximum length of the text. The non-parametric queries are predicted with an MLP from the 256 visual tokens with the highest scores. Besides, the number of layers in the decoder is set to $N_E = 6$. The decoder predicts object features $\mathbf{O} \in \mathbb{R}^{k \times d}$, where $k = 256$ is the number of candidate objects, and $d = 288$ is the feature dimension.

**Performance in the process of BUTD-DETR simplification to L3Det**    Table 7 shows the changes in the visual grounding performance from top to bottom during the process of simplifying BUTD-DETR to our L3Det. From the results, it can be seen that by directly inputting text tokens and object queries parallelly into the decoder, it can compensate for the performance drop caused by abandoning the cross-encoder and even achieve better performance ($51.0 \to 47.2 \to 51.3$). Besides, using alignment loss following GLIP can further improve the model's performance to 52.8 ($> 52.2$, the performance of BUTD-DETR) while not using box stream compared with BUTD-DETR.

**Comparision with existing detection methods for L3Det**    To demonstrate our L3Det's strong detection capability, we directly train L3Det on ScanNet 18 classes using the 18 categories combination detection prompt, and the experiments in Table 6 show L3Det achieves higher detection performance.

Table 6: 18 class 3D object detection results on ScanNetV2.

| Method | $mAP_{50}$ |
|---|---|
| 3DETR | 44.6 |
| GroupFree3D | 48.9 |
| L3Det | 50.1 |

**Combining Object2Scene with existing methods**  Here we attempt to use Object2Scene to enable the close-set 3D object detector to obtain open-vocabulary detection capability. Since both GroupFree3D and 3DETR are close-set 3D object detectors and do not possess the text input capability, we modified their class prediction to 20 classes, which cover all the categories in OV-ScanNet20 benchmark, but only seen annotations (10 classes) are used for training in the actual training process. Then we introduce unseen objects using Object2Scene to expand the training dataset. Results on OV-ScanNet20 in Table 7 show our Object2Scene is general, and L3Det is also better than GroupFree3D and 3DETR due to the text prompt input ability and architecture advantages.

Table 7: Detection results on unseen classes of OV-ScanNet20.

| Method | $mAP_{25}$ |
|---|---|
| 3DETR | 1.31 |
| GroupFree3D | 0.53 |
| 3DETR + Object2Scene | 14.23 |
| GroupFree3D + Object2Scene | 15.16 |
| L3Det + Object2Scene | 23.98 |

**Alignment Matrix Generation for L3Det**  In the second paragraph of Section 4.1 of the main paper, we introduce the training supervision for L3Det, where calculating the alignment loss requires a target alignment score matrix $\mathbf{S}_{target} \in \{0, 1\}^{N \times M}$. The key to generating the target alignment score matrix is the fine-level alignment between the text tokens and 3D boxes which is typically not provided in most of visual grounding datasets including ScanRefer (Chen et al., 2020). We use the off-the-shelf tool following EDA (Wu et al., 2022) to parse the text description, generate the grammatical dependency trees, and obtain the position label. For example, given a sentence *"It is a white table. It is next to a backboard"* consisting of multiple objects, the main object in this sentence is *"table"* and the corresponding position label is *"0000100...."*.

**Training Details**  The code is implemented based on PyTorch. Our model is trained on two NVIDIA A100 GPUs with a batch size of 24. We freeze the pretrained text encoder and use a learning rate of $1 \times 10^{-3}$ for the visual encoder and a learning rate of $1 \times 10^{-4}$ for all other layers in the network. It takes around 25 minutes to train an epoch, and our model is trained for 120 epochs. The best model is selected based on the performance of the validation set.

## B  VISUAL GROUNDING RESULTS

Our proposed L3Det model unifies the 3D object grounding and detection with the same framework, and we report the language-based 3D grounding performance trained on ScanRefer (Chen et al., 2020) in Table 8. Compared with previous works such as BUTD-DETR (Jain et al., 2022), our proposed L3Det achieves better grounding results with a simpler model architecture. We hope our proposed L3Det will serve as a new 3D grounding and detection baseline for its simple, effective, and unified model architecture.

## C  QUALITATIVE ANALYSIS

In this section, we illustrate more scenes generated by our Object2Scene approach in Figure 7 and more visualization results in Figure 8. Figure 7 shows several scenes generated by Object2Scene, where the 3D objects are inserted into the real-scanned scenes in a reasonable manner. As illustrated in Figure 8, L3Det can locate all objects belonging to the category described in the input text prompt

Table 8: Performance comparisons on language grounding on ScanRefer (Chen et al., 2020)

| Method | Unique@0.25 | Unique@0.5 | Multi@0.25 | Multi@0.5 | Overall@0.25 | Overall@0.5 |
|---|---|---|---|---|---|---|
| ReferIt3DNet (Achlioptas et al., 2020) | 53.8 | 37.5 | 21.0 | 12.8 | 26.4 | 16.9 |
| ScanRefer (Chen et al., 2020) | 63.0 | 40.0 | 28.9 | 18.2 | 35.5 | 22.4 |
| TGNN (Huang et al., 2021) | 68.6 | 56.8 | 29.8 | 23.2 | 37.4 | 29.7 |
| IntanceRefer (Yuan et al., 2021) | 77.5 | 66.8 | 31.3 | 24.8 | 40.2 | 32.9 |
| FFL-3DOG (Feng et al., 2021) | 78.8 | 67.9 | 35.2 | 25.7 | 41.3 | 34.0 |
| 3DVG-Transformer (Zhao et al., 2021) | 77.2 | 58.5 | 38.4 | 28.7 | 45.9 | 34.5 |
| SAT-2D (Yang et al., 2021) | - | - | - | - | 44.5 | 30.1 |
| BUTD-DETR (Jain et al., 2022) | 84.2 | 66.3 | 46.6 | 35.1 | 52.2 | 39.8 |
| L3Det | **84.8** | **67.1** | **47.1** | **35.9** | **52.8** | **40.2** |

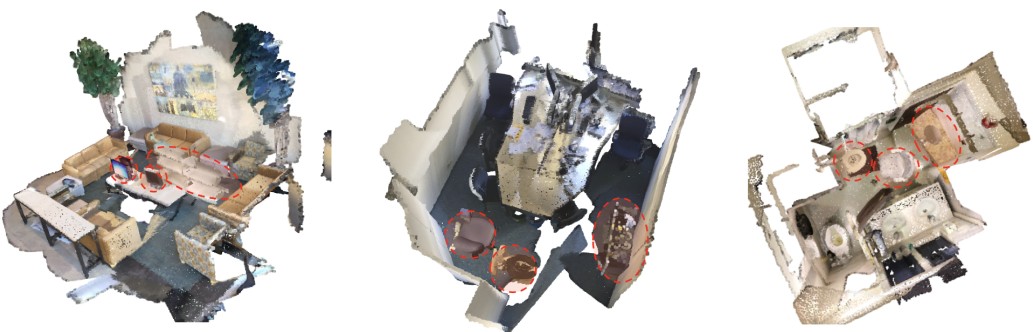

Figure 7: Sample scenes generated by Object2Scene. The objects surrounded by the red circle in the figure are sampled from 3D object datasets and inserted into the real-scanned scene.

covering various object sizes. Nevertheless, we find that our model may sometimes incorrectly detect the objects (illustrated in the top middle sub-figure in Figure 8) or miss the objects. For example, if the chairs are tucked under the table, the actual point cloud distribution of the chairs and the point cloud distribution of chairs we insert into by Object2Scene are often very different, making it difficult to detect. Those failure cases might be due to the distribution misalignment between the scanned point cloud in the scene and the point cloud of the inserted objects from other datasets. We leave this issue for future work.

Table 9: Ablation Study.

(a) Performance of different training epochs when 40% of the 3D objects from the 3D object dataset are used for training.

| Training Epochs | $mAP_{25}$ |
|---|---|
| 30 | 11.87 |
| 45 | 15.43 |
| 60 | 18.99 |
| 100 | 20.62 |
| 120 | 21.31 |

(b) Performance of different data ratio used in Object2Scene with 120 training epochs. Data ratio refers to the ratio of objects from the 3D object dataset that are used for training. It reflects the diversity of 3D objects that are inserted to the scenes.

| Data Ratio | $mAP_{25}$ |
|---|---|
| 40% | 12.56 |
| 80% | 18.11 |
| 100% | 21.31 |

# D  ABLATION STUDY

In this section, we provide more ablation studies on how to use the data generated by Object2Scene for training. During training, the Object2Scene approach generates augmented scenes online, *i.e.*, the inserted objects and locations to insert objects are sampled at each iteration. We investigate two factors: 1) the number of training epochs, and 2) the diversity of inserted objects, *i.e.*. the ratio of data from the 3D object dataset that is used for training. Table 9a demonstrates that more training epochs lead to better performance. Table 9b indicates that increasing the diversity of inserted 3D objects improves the performance.

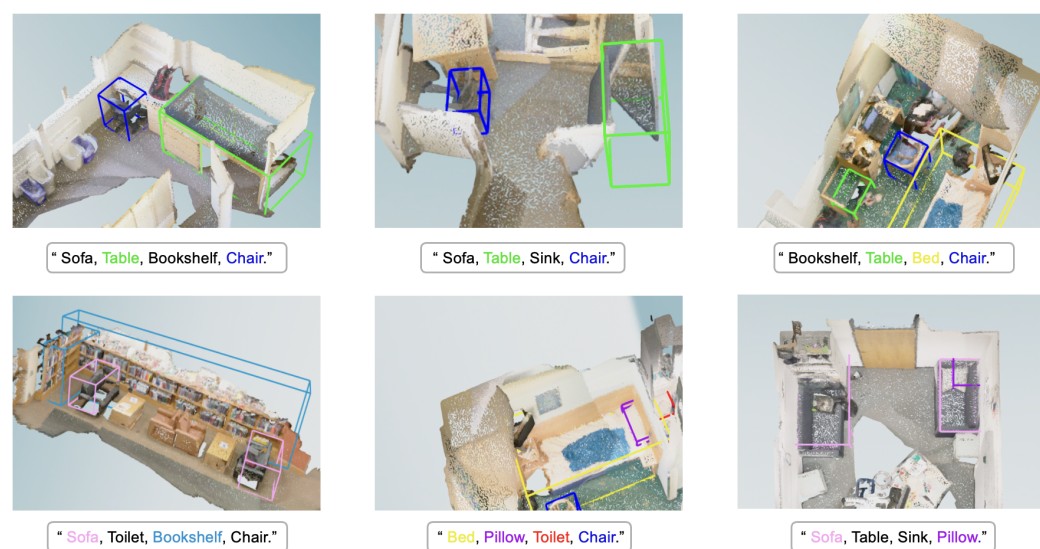

Figure 8: Qualitative results for open-vocabulary 3D object detection results. For each scene, the detection prompt is shown under the input point cloud. The colors of bounding boxes correspond to the classes in the prompts.

## E    TRANSFERABILITY TO NEW DATASETS

We explore the transferability of our 3D detectors by evaluating the cross-dataset transfer performances between OV-ScanNet20 and OV-SUN RGB-D20. The transferability of our 3D detector mainly comes from the pretrained text encoder and the robust and transferable 3D feature representations trained with objects from multiple source datasets using the cross-domain category-level contrastive loss. The object detector trained on OV-ScanNet20 achieves an $mAP_{25}$ of 16.34% when tested on OV-SUN RGB-D20 dataset, and the object detector trained on OV-SUN RGB-D20 achives an $mAP_{25}$ of 17.11% when tested on OV-ScanNet20, demonstrating the transferability of the object detectors trained with Object2Scene.

