# OpenReview forum: "Object2Scene: Putting Objects in Context for Open-Vocabulary 3D Detection"
_ICLR.cc/2024/Conference — Submitted to ICLR 2024_

### Official Review · Reviewer_ciKG · 2023-10-25

**Soundness:** 2 fair
**Presentation:** 3 good
**Contribution:** 2 fair
**Rating:** 5
**Confidence:** 5

**Summary:**

This paper studies open-vocabulary 3D object detection. They propose Object2Scene, the first approach that leverages large-scale large-vocabulary 3D object datasets to augment existing 3D scene datasets for open-vocabulary 3D object detection. Object2Scene inserts objects from different sources into 3D scenes to enrich the vocabulary of 3D scene datasets and generates text descriptions for the newly inserted objects. They further introduce a framework that unifies 3D detection and visual grounding, named L3Det, and propose a cross-domain category-level contrastive learning approach to mitigate the domain gap between 3D objects from different datasets. Extensive experiments on existing open-vocabulary 3D object detection benchmarks show that Object2Scene obtains superior performance over existing methods.

**Strengths:**

- This paper is the first paper that leverages large-vocabulary 3D object datasets to augment the existing 3D scene datasets.

- They propose a contrastive learning technique to mitigate the cross-dataset domain gaps.

- They propose several new open vocabulary 3D detection benchmarks.

- Their method outperforms the existing baseline approaches on the benchmarks, which demonstrates the effectiveness of their proposed method.

- The paper writing is clear and easy to follow.

**Weaknesses:**

- The main idea of this paper is essentially an object cut-paste augmentation, which has been exploited as a common practice in most 3D object detectors. For example, starting from Point R-CNN, those 3D object detectors leverage object cut-paste to augment the 3D driving scenes. The proposed object anchor selection, normalization, and placement share very similar spirits with Point R-CNN [1] and Back-to-Reality [2].

- The contrastive learning strategy has also been studied by many relevant papers. For example, CLIP^2 [3] leverages contrastive learning to align the text and point cloud representations for open-vocabulary 3D object detection.

- Text prompts are generated by pre-defined rules and lack generalization ability.

- The proposed detection framework didn't change much compared to the existing detectors. The detector is trained with supervised learning and doesn't rely on open-vocabulary foundation models such as CLIP. Hence it is hard to say whether the proposed method has the open-vocabulary detection ability, as the detection vocabulary is actually constrained by the 3D object datasets.

- Evaluation is also questionable: The authors split the datasets into seen and unseen categories, but those unseen categories are not really ``unseen", since the 3D object datasets they used can have those categories. Hence it is an unfair comparison with the baseline methods.

- Essentially this paper is not an open-vocabulary 3D detection paper. It is 3D detection with categories augmentation by inserting new objects from 3D object datasets. The detected categories are bounded by the used object datasets.

[1] Shi et al. PointRCNN: 3D Object Proposal Generation and Detection from Point Cloud. CVPR 2019.

[2] Xu et al.  Back to reality: Weakly supervised 3D object detection with shape-guided label enhancement. CVPR 2022.

[3] Zeng et al. CLIP$^2$: Contrastive Language-Image-Point Pretraining from Real-World Point Cloud Data. CVPR 2023.

**Questions:**

Please refer to the weaknesses.

---

> ### Author Response · Authors · 2023-11-20
>
> **1. The main idea of this paper is essentially an object cut-paste augmentation, which has been exploited as a common practice in most 3D object detectors. For example, starting from Point R-CNN, those 3D object detectors leverage object cut-paste to augment the 3D driving scenes. The proposed object anchor selection, normalization, and placement share very similar spirits with Point R-CNN [1] and Back-to-Reality [2].**
>
> Thanks for your reminder, we clarify the differences with the existing data augmentation methods in the overall rebuttal section and have updated the related discussion in the revised paper.
>
> **2. The contrastive learning strategy has also been studied by many relevant papers. For example, CLIP^2 [3] leverages contrastive learning to align the text and point cloud representations for open-vocabulary 3D object detection.**
>
> Contrastive learning itself is a very versatile learning strategy, widely used in a variety of tasks. However, how to design suitable contrastive learning objectives and construct appropriate positive and negative learning samples for different scenarios is an important issue. In our paper, we address the domain gap problem introduced by multi-object dataset training through a carefully designed category-level contrastive learning loss. However, in CLIP^2, contrastive learning is employed to align text and point cloud representations.
>
> **3. Text prompts are generated by pre-defined rules and lack generalization ability.**
>
> Our generated prompts are indeed rule-based and to some extent lack generative ability. However, the purpose of generating prompts is to align the point clouds and texts of objects in scenes, thereby achieving open-vocabulary. Experimental results validate the effectiveness of our prompt generation. Of course, we believe that more complex and natural language descriptions will have better generalization capabilities and bring improved point cloud-text alignment results. We will consider introducing LLM or some other tools in the future to generate more natural prompts for experiments.
>
> **4. The proposed detection framework didn't change much compared to the existing detectors. The detector is trained with supervised learning and doesn't rely on open-vocabulary foundation models such as CLIP. Hence it is hard to say whether the proposed method has the open-vocabulary detection ability, as the detection vocabulary is actually constrained by the 3D object datasets.**
>
> We offer the related statement in the overall rebuttal section. Thanks for your question.
>
> **5. Evaluation is also questionable: The authors split the datasets into seen and unseen categories, but those unseen categories are not really ``unseen", since the 3D object datasets they used can have those categories. Hence it is an unfair comparison with the baseline methods.**
>
> We offer the related statement in the overall rebuttal section.
>
> **6. Essentially this paper is not an open-vocabulary 3D detection paper. It is 3D detection with categories augmentation by inserting new objects from 3D object datasets. The detected categories are bounded by the used object datasets.**
>
> We restate the definition of open-vocabulary 3D detection in the overall rebuttal section.

---

> > ### Author Response · Authors · 2023-11-21
> >
> > Thanks for your review, may I know whether your problems have been solved after the rebuttal? Or are there any other questions or confusions?

---

### Official Review · Reviewer_UJPz · 2023-10-26

**Soundness:** 3 good
**Presentation:** 3 good
**Contribution:** 3 good
**Rating:** 6
**Confidence:** 5

**Summary:**

This paper studies open-vocabulary 3D object detection and proposes object2scene. They propose to use large-scale 3D object datasets and insert objects from different sources into 3D scenes. They also generate text descriptions for the newly inserted objects to distinguish from unseen category objects from the original scene. In obtaining the final open-vocabulary 3D model, they unify 3D detection and visual grounding and propose a cross-domain object-level contrastive learning approach to mitigate the domain gap between 3D objects from different datasets.

**Strengths:**

1. The method proposed is also clearly described and easy to understand.

2. The authors try to use existing 3D object datasets and combine them with the scene dataset. Based on this, they unify the 3D object detection task and the visual grounding task to realize the direct implementation of open-vocabulary detection models in 3D space.

3. The experiments are comprehensive, they outperform other open-vocabulary detectors.

**Weaknesses:**

I have reviewed this paper before. Compared to the last version, some presentation problems have been well-addressed in the version and it is now smoother in logic and easier to follow.

Previously, it was more like a dataset paper, but this time, it looks much better.

May I ask several questions:
1. Compared to the last version, the results in Table 1 are slightly dropped, why?
2. May I know the scale of inserted 3D objects and generated prompts? For example, the number of them? Since contrastive training like GLIP usually requires large-scale objects and prompts.
3. As mentioned in the last rebuttal, "we will add and discuss the cut and mix augmentation methods, such PointCutMix, in our final version." I could not find the discussion.
4. Similarly, it is mentioned that "we will consider using 'spatial proximity' instead of 'affordance' and update in our final paper." Do you have any comment about this?

**Questions:**

No further questions.

---

> ### Author Response · Authors · 2023-11-20
>
> **1. Compared to the last version, the results in Table 1 are slightly dropped, why?**
>
> Since in the former version, there exist the parf of dataset overlap between the test set ScanNet and the training dataset ScanObjectN, so in the last version we change the ScanObjectNN to OmniObject3D, a new large-scale real-scanned 3D object dataetst, which has no overlap with ScanNet.
>
> **2. May I know the scale of inserted 3D objects and generated prompts? For example, the number of them? Since contrastive training like GLIP usually requires large-scale objects and prompts.**
>
> For OV-ScanNet20 and OV-SUN RGBD20 benchmark , the inserted object datasets cover 6w 3D objects with 190 categories. As for the generated prompts, since they are generated randomly and dynamically, based on our statistics, the actual number of prompts generated during the training process is approximately 500,000.
>
> **3. As mentioned in the last rebuttal, "we will add and discuss the cut and mix augmentation methods, such PointCutMix, in our final version." I could not find the discussion.**
>
> Thanks for your reminder, we have added PointCutMix in the related work section in our updated version.
>
> **4. Similarly, it is mentioned that "we will consider using 'spatial proximity' instead of 'affordance' and update in our final paper." Do you have any comment about this?**
>
> Actually we have updated most of our descriptions in this ICLR version, thanks for your suggestions in last review. In this version, after double check, the statement about 'affordance' only lies in the " Specifically, we choose seen objects in the scene as reference objects to guide the physically reasonable 3D object insertion according to their physical affordance. ", and we have revised it in our updated version. Besides, in this version, we clarify the generated prompts into three categories: Vertical Proximity, Horizontal Proximity and Allocentric in Section 3.2, and more details can be found in our supplementary.

---

> > ### Comment · Reviewer_UJPz · 2023-11-20
> >
> > Thank you for the response. I think the authors have addressed my concerns. After going through the responses to other reviewers, I would like to keep my score.

---

### Official Review · Reviewer_sdvC · 2023-11-01

**Soundness:** 2 fair
**Presentation:** 2 fair
**Contribution:** 2 fair
**Rating:** 3
**Confidence:** 4

**Summary:**

This paper proposes a technique to improve 3D object detection, by creating additional synthetic data in the form of randomized rooms with language grounding. The paper also contributes a new architecture for the task, which takes text and a pointcloud as input and produces 3D bounding boxes as output. The method also uses a contrastive loss to help reduce inter-class dissimilarity and increase intra-class dissimilarity, which is applied on the features within the decoder of the model.

**Strengths:**

This paper makes a convincing case for creating additional diverse data with language grounding labels to train open-vocabulary 3d detectors, and the method presented here to create such data seems like a reasonable one.

**Weaknesses:**

This paper never brings up the highly overlapping work "RandomRooms: Unsupervised Pre-training from Synthetic Shapes and Randomized Layouts for 3D Object Detection" from ICCV 2021. That paper covers the exact same idea of placing random objects into random locations to create diverse training rooms. To me it seems critical to discuss this work, and differentiate from it in some concrete ways.

The paper says "With the proposed cross-domain category-level contrastive loss, the model is forced to learn generalizable feature representations based on their class labels irrespective of the source dataset of the objects." It is unclear to me why the contrastive loss will force generalizable features. I think the high-level idea here is that the contrastive loss will make the features more similar within a class, but this effect is already achieved by the classification loss. Although empirically the contrastive loss may help further still, I don't think the given explanation really works.

Section 3.2 is very difficult to follow. I am lost from the first sentence: "The 3D object insertion approach composes new 3D scenes with original scenes annotated by seen categories and new objects from large vocabulary categories covering seen and unseen categories." I think a comma will be good somewhere, or maybe this sentence can be broken into multiple sentences. The notation <target - class><spatial - relationship><anchor - class> is confusing for me. If these are 2-tuples, what is a "spatial" and what is a "relationship"? Finally this section states that the "Absolute Location Prompt" is proposed as a mechanism to validate the "Relative Location Prompt", but it is unclear how this mechanism works. The end of that paragraph is confusing as well, as it seems to say (via double negatives) that the prompts might be less unclear than the relative ones ("such prompts are limited by fewer ways of expression and unclear descriptions which may not be discriminative").

**Questions:**

direclty -> directly

---

> ### Author Response · Authors · 2023-11-20
>
> **1.This paper never brings up the highly overlapping work "RandomRooms: Unsupervised Pre-training from Synthetic Shapes and Randomized Layouts for 3D Object Detection" from ICCV 2021. That paper covers the exact same idea of placing random objects into random locations to create diverse training rooms. To me it seems critical to discuss this work, and differentiate from it in some concrete ways.**
>
> Thanks for your reminder! We have added RandomRooms to our related work in our updated version for discussion. Here we want to clarify that the only common point between Object2Scene and RandomRooms may be that they both use 3D object datasets to enhance 3D scene understanding. However, they have significant differences:
> 1. Completely different implementation methods: RandomRooms proposes to generate random layouts of a scene and construct the virtual scene using objects from synthetic CAD datasets, learning the 3D scene representation. On the other hand, Object2Scene inserts 3D objects into real-scanned 3D scenes to enhance diversity in existing scenes and further guides detectors with open-vocabulary detection capability through appropriate prompts.
> 2. Different problems addressed: RandomRooms aims to pretrain models using generated virtual scenes, which can serve as better initialization for later fine-tuning on 3D object detection tasks. Object2Scene aims to train a model that can be directly used in real-scanned scenes and conduct 3D understanding.
> 3. In initial experiments, we attempted training by directly composing virtual scenes using RandomRooms approach. However, due to the significant domain gap between constructed virtual scenes and real-scanned scenes, the detector trained on virtual scenes only achieved poor performance in open-vocabulary detection in real-scanned scenarios. By adopting our proposed Object2Scene, which is a less invasive method, we can avoid the scene-level domain gap and make it more practical.
>
> **2.The paper says "With the proposed cross-domain category-level contrastive loss, the model is forced to learn generalizable feature representations based on their class labels irrespective of the source dataset of the objects." It is unclear to me why the contrastive loss will force generalizable features. I think the high-level idea here is that the contrastive loss will make the features more similar within a class, but this effect is already achieved by the classification loss. Although empirically the contrastive loss may help further still, I don't think the given explanation really works."**
>
> Thanks for your question, here we show more experiment results to support our claim. Table 4 in our paper demonstrates the effectiveness of our proposed cross-domain category-level contrastive loss. It is shown that cross-domain object-level contrastive learning boosts the mAP25 of open-vocabulary 3D detection from 7.34% to 11.21%.  Besides, we also offer TSNE visualization to support our interpretation. The figure can be found in our updated version.
>
> **3. Section 3.2 is very difficult to follow. I am lost from the first sentence: "The 3D object insertion approach composes new 3D scenes with original scenes annotated by seen categories and new objects from large vocabulary categories covering seen and unseen categories." I think a comma will be good somewhere, or maybe this sentence can be broken into multiple sentences. The notation <target - class><spatial - relationship><anchor - class> is confusing for me. If these are 2-tuples, what is a "spatial" and what is a "relationship"? Finally this section states that the "Absolute Location Prompt" is proposed as a mechanism to validate the "Relative Location Prompt", but it is unclear how this mechanism works. The end of that paragraph is confusing as well, as it seems to say (via double negatives) that the prompts might be less unclear than the relative ones ("such prompts are limited by fewer ways of expression and unclear descriptions which may not be discriminative").**
>
> 1. The first sentence in Setection 3.2 has been updated to "The 3D object insertion approach creates new 3D scenes by combining original scenes that have been annotated with seen categories, along with new objects from large vocabulary categories that include seen and unseen ones." in our uptaded version.
> 2. The notation has been updated to " <target> <spatial relationship> <anchor>" in our updated version.
> 3. As for section states about "Absolute Location Prompt", this paragraph has been rewritten to be more clear:
>  [Besides, we also propose the Absolute Location Prompt. The construction of this type of prompt relies solely on the object's position in the scene, and not on other objects in the scene. For example: "a table that is closer to the center/corner/wall of the room". Compared to the Relative Location Prompt, the prompt is simpler, but it offers a cruder positioning of the object.]

---

> > ### Author Response · Authors · 2023-11-21
> >
> > Thanks for your review, may I know whether your problems have been solved after the rebuttal? Or are there any other questions or confusions?

---

### Official Review · Reviewer_XyDn · 2023-11-01

**Soundness:** 3 good
**Presentation:** 2 fair
**Contribution:** 2 fair
**Rating:** 6
**Confidence:** 4

**Summary:**

This paper introduces Object2Secene, a method that samples objects from large-scale 3D object datasets and integrates them into 3D scenes. This process is used to train open-vocabulary 3D object detection models. The authors generate language grounding prompts to facilitate the learning of inserted objects. They also implement a cross-domain category-level contrastive learning to mitigate the domain gap between inserted 3D objects and 3D scenes.

**Strengths:**

1. Object2scene enhances the vocabulary diversity of objects in original 3D scene data by inserting extra 3D objects, which is straightforward and does not rely on large-scale image-text datasets or pre-trained 2D open-vocabulary detection models.
2. Their proposed method significantly outperforms previous methods.

**Weaknesses:**

1.	Do the inserted object classes fully contain the unseen classes in the test set? If the unseen classes are fully covered during training, does this still counts as open vocabulary 3D object detection. Moreover, if these unseen classes are not included during object insertion, how well would the trained model perform on the unseen classes?
2.	While using existing 3D object datasets can provide annotated objects across more categories for model training, the training process may still be limited by the vocabulary of these datasets. Learn semantics beyond these datasets’ object categories pose a challenge for the model.
3.	The “Object Placement” process is central to Object2Scene but needs further clarification. How are injection regions determined? How is physical reasonability checked during placement? A pseudo code on this process could enhance understanding.
4.	Lack of comparison on the larger OV-ScanNet200 benchmark.

**Questions:**

Please see the weakness part.

---

> ### Author Response · Authors · 2023-11-20
>
> **1. Do the inserted object classes fully contain the unseen classes in the test set? If the unseen classes are fully covered during training, does this still counts as open vocabulary 3D object detection. Moreover, if these unseen classes are not included during object insertion, how well would the trained model perform on the unseen classes?**
>
> In our OV-ScanNet20 and OV-RGB-D20 benchmarks, inserted object classes fully contain the unseen classes in the test set, but not in OV-ScanNet200 benchmark.  If the unseen classes are fully covered during training, this still counts as open vocabulary 3D object detection according to the definition of open-vocabulary object detection stated in overall rebuttal statement, and here we offer the part of classes which are not included in training datasets:
> |  | copier | Computer tower | ottoman | soap  |
> | --- | --- | --- | --- | --- |
> | L3Det | 0.342 | 0.12 | 1.23 | 0.00 |
>
> **2.While using existing 3D object datasets can provide annotated objects across more categories for model training, the training process may still be limited by the vocabulary of these datasets. Learn semantics beyond these datasets’ object categories pose a challenge for the model.**
>
> Yes, it's true that learning semantics beyond these datasets’ object categories poses a challenge for the model during the training. However, once the number of object categories included in the training data is large enough, like Detic using ImageNet21K datasets to broaden the object concept, then the model's open-vocabulary capability can cover most objects in daily scenes.
>
> **3.The “Object Placement” process is central to Object2Scene but needs further clarification. How are injection regions determined? How is physical reasonability checked during placement? A pseudo code on this process could enhance understanding.**
>
> Thanks for your advice, we have added this pseudo code in the appendix of our updated paper for reference.
>
> **4.Lack of comparison on the larger OV-ScanNet200 benchmark.**
>
> |  | head |  common | tail |
> | --- | --- | --- | --- |
> | OV-PointCLIP | 2.76 | 0.56 | 0.0 |
> | Detic-ModelNet | 1.56 | 0.12 | 0.0 |
> | L3Det | 13.1 | 10.1 | 3.4 |

---

> > ### Comment · Reviewer_XyDn · 2023-11-22
> >
> > Thank you for addressing my questions. While it can be expected that augmenting objects in 3D scenes could improve detection performance for related categories, this improvement may not be significant enough to warrant substantial progress in open-vocabulary object detection specifically. Hence, I prefer to keep my initial score.

---

### Author Response · Authors · 2023-11-20
**Overall Rebuttal Section for some problems**

Dear reviewers,
We thank all the reviewers for your insightful comments. We are encouraged that you found our method well-motivated (reviewer XyDn), clear and effective (reviewer UJPz), and with impressive results (reviewer XyDn, UJPz). We address all the questions as below and update the corresponding results. After analyzing your comments, we have identified some common issues. Therefore, we will begin by addressing three key points:
1. **The task definition of open-vocabulary object detection:**  Here we reintroduce the definition of open-vocabulary object detection. [1] first introduce open-vocabulary object detection, which aims to detect and localize objects for which no bounding box annotation is seen during training. Different from zero-shot object detection, the objects of unseen categories not from training datasets are allowed to be utilized during training, which can refer to [2]. Detic uses the ImageNet21K datasets to transfer more object concept knowledge from ImageNet21K through sharing the Classification Head.
2. **Detection vocabulary restriction problem:**  This is a good challenge proposed by the reviewer XyDn and ciKG, indeed the model  typically only achieves promising detection results on the classes included in training datasets. However, this challenge also lies in 2D domain. The main difference is that 2D training datasets typically have a much larger number of object classes compared to 3D point clouds. Besides, with the rise of large-scale 3D object datasets like Objaverse (including 818k objects and 21k categories), our method can fully expand the vocabulary size and achieve real open-world object detection ability.
3. **Comparison with the existing data augmentation methods:** Here we discuss the differences between Object2Scene and other data augmentation methods in the following table:

|  | Motivation  | Techincally Desgin | Scene | Language  Generation  |
| --- | --- | --- | --- | --- |
| PointCutMix | To achieve robust object-level point cloud classification  | It finds the optimal assignment between two point clouds and generates new training data by replacing the points in one sample with their optimal assigned pairs. |  Indoor  |  no |
| Copy-paste in PointRCNN | To create more diverse training samples and enhance close-set detection performance | It copies the 3D objects and pastes them into the scene data | Outdoor ( 3D driving scenes) | no |
| RandomRooms | Due to the lack of real-scanned scene datasets, synthetic 3D object datasets are used to enhance close-set detection performance | It generates random layouts of a scene and construct the virtual scene using objects from synthetic CAD datasets. |  Indoor |  no |
| BackToReality | Due to the lack of real-scanned scene datasets, synthetic 3D object datasets are used to enhance close-set detection performance | It relies on position-level annotation of real-scene and then constructs the physically reasonable virtual scene to guide the 3D detector training |  Indoor |  no |
| Ours | Due to the lack of vocabulary size in 3D scene, We introduce large vocabulary 3D object datasets to enrich the open-vocabulary detecion ability. | Our object placement is specially designed for prompt generation in more complex and diverse indoor scene, meanwhile, to make the inserted object physically reasonable and conform to the real scene, we propose the anchor-guided object insertion approach. | Indoor  | yes |

[1] Open-Vocabulary Object Detection Using Captions
[2] Detecting Twenty-thousand Classes using Image-level Supervision

---

### Meta-Review · Area_Chair_jQC8 · 2023-12-10

**Metareview:**

**Summary**
This paper proposes to tackle open-vocabulary 3D object detection in point clouds by inserting objects from 3D object datasets to augment 3D scene datasets.  Objects are selected based on predefined list of similar object categories so that the number of points and overall object size is normalized.  After appropriate objects are selected, they are placed in the scene relative to other objects so that even if there are existing objects of the same category in the scene, the reference to the object will be unique.  The paper proposes a transformer-based model, L3Det, that attempts to unify 3D visual grounding and 3D object detection where the objects to detect are fed into a pretrained text encoder (RoBERTa) and the model is trained to detect object corresponding to the input text.  Experiments are conducted on ScanNet20, SUN-RGBD20, and ScanNet200 with seen and unseen classes.

The paper indicates that main contributions of the work are:
- Approach to generate 3D data for open-vocabulary 3D object detection by inserting 3D objects into 3D scene dataset
- Language grounded 3D detection framework L3Det with cross-domain category-level contrastive loss
- Experiments on 3D object detection benchmarks on ScanNet and SUN RGB-D

**Strengths**
- The idea of augmenting 3D scene datasets by inserting objects is simple and intuitive
- Experiments show the proposed method outperforms prior work

**Weaknesses**
- Reviewers had questions as to whether this work is truly "open-vocabulary" or more using data-augmentation for 3D object detection (ciKG,XyDn).
- Ability to handle open-vocabulary is limited by the types of objects that can be inserted (XyDn)  There is not enough discussion of this in the paper.
- Other reviewers found the parts of the paper confusing and hard to follow (sdvC)
- Some important points are not well described
  - For instance, whether the object inserted covers the "unseen" classes or not (XyDn) and if it covers the "unseen" classes, what does it mean for a class to be "unseen"?
  - Description of object placement is missing some details  (XyDn)
- Limited discussion of relevant work on open-vocabulary 3D object detection.  The 3D referential language grounding is also missing discussion of relevant work such as ScanRefer [Chen et al. 2020], 3DRefTransformer [Abdelreheem et al. 2022], UniT3D [Chen et al. 2023], etc.

**Recommendation**
Due to issues pointed out by reviewers, the AC feels the paper is not ready for acceptance at ICLR and recommends reject.

**Suggestions for improvement**
1. Determine the focus of the work - is it "open-vocabulary" object detection or (or actually "open-class") object detection (see note below), or is the focus the insertion of objects for data-augmentation?
- For "open-vocabulary", the AC recommends showing experiments with a large vocabulary size and clarifying the vocabulary used in training vs test.
- For "open-class", details should be provided how many of the "unseen" classes are covered by the object insertion strategy and include discussion of the limitations of the proposed method.
- If the focus is more on the data-augmentation, there should be experiments that compare against different augmentation strategies and discussing those in detail in the paper.
2. Improve the writing to clarify important points and correct typos:
 "reprenstation" => "representation"
 "categoreis" => "categories"
 "IMPLEMNTATION" => "IMPLEMENTATION"
 "direclty" -> "directly"

Note about "open-vocabulary" - one concern brought up by reviewers is whether the work is actually "open-vocabulary" . According to the paper, "open-vocabulary object detection aims at detecting the categories that are not provided bounding box labels during training". To the best of the AC's knowledge, this is not an accurate description of "open-vocabulary", but rather of "open-world" or "open-class" object detection.  The distinction between "open-vocabulary" and "open-world" is that "open-vocabulary" is more about the names / words (e.g. vocabulary) used to refer to the classes and not just the classes themselves.  For instance, in "open-vocabulary", you may have a class "couch" that is seen during training, but during test time, you may also need to correctly identify objects that are called "sofa".  This is different from "open-world".  Typically, you would expect "open-vocabulary" models to handle "open-world" cases as well, and to test this aspect of "open-vocabulary" systems, it would be common to have the test set contain classes that are not seen during training.

**Justification For Why Not Higher Score:**

Reviewers had questions about the work actually tackles "open-vocabulary" object detection or is it more 3D detection with categories augmentation.  As the focus of the work is unclear, it is hard for reviewers to judge whether appropriate experiments and claims have been made. The AC feels in it's current state, the paper is confusing on several points and is not quite ready for publication.

**Justification For Why Not Lower Score:**

N/A

---

### Decision · Program_Chairs · 2024-01-16

Reject